# Orally Administered Natural Lipid Nanoparticle-Loaded 6-Shogaol Shapes the Anti-Inflammatory Microbiota and Metabolome

**DOI:** 10.3390/pharmaceutics13091355

**Published:** 2021-08-28

**Authors:** Chunhua Yang, Dingpei Long, Junsik Sung, Zahra Alghoul, Didier Merlin

**Affiliations:** 1Digestive Disease Research Group, Institute for Biomedical Sciences, Georgia State University, Atlanta, GA 30303, USA; dlong26@gsu.edu (D.L.); jsung9@gsu.edu (J.S.); zalghoul@gsu.edu (Z.A.); dmerlin@gsu.edu (D.M.); 2Atlanta Veterans Affairs Medical Center, Decatur, GA 30302, USA; 3Department of Chemistry, Georgia State University, Atlanta, GA 30303, USA

**Keywords:** natural-lipid nanoparticles, 6-shogaol, fecal metabolites, microbiota compositional change, inflammation, ulcerative colitis

## Abstract

The past decade has seen increasing interest in microbiota-targeting therapeutic strategies that aim to modulate the gut microbiota’s composition and/or function to treat chronic diseases, such as inflammatory bowel disease (IBD), metabolic symptoms, and obesity. While targeting the gut microbiota is an innovative means for treating IBD, it typically requires an extended treatment time, hampering its potential application. Herein, using an established natural-lipid nanoparticle (nLNP) platform, we demonstrate that nLNPs encapsulated with the drug candidate 6-shogaol (6S/nLNP) distinctly altered microbiota composition within one day of treatment, significantly accelerating a process that usually requires five days using free 6-shogaol (6S). In addition, the change in the composition of the microbiota induced by five-day treatment with 6S/nLNP was maintained for at least 15 days (from day five to day 20). The consequent alteration in the fecal metabolic profile stemming from this compositional change manifested as functional changes that enhanced the in vitro anti-inflammatory and wound-healing efficacy of macrophage cells (Raw 264.7) and epithelial cells (Caco-2 BBE1), respectively. Further, this metabolic compositional change, as reflected in an altered metabolic profile, promoted a robust anti-inflammatory effect in a DSS-induced mouse model of acute colitis. Our study demonstrates that, by near-instantly modulating microbiota composition and function, an nLNP-based drug-delivery platform might be a powerful tool for treating ulcerative colitis.

## 1. Introduction

Ulcerative colitis (UC) is the most common form of inflammatory bowel disease (IBD) [1], which affects more than 1 M patients in the US and 5 M worldwide [2]. UC-caused colonic ulcer and inflammation can cause irregularities in the turnover of cells in the intestinal lining, increasing the chances of inducing colorectal cancer [3,4]. Thus, timely treatment of the UC is crucial for preventing the induction of UC-associated diseases [3,5]. Although traditional treatment of UC (including anti-inflammatory drugs or antibiotics) has an immediate and direct effect, patients involved with such treatments often suffer from severe systemic side effects [6].

Recent discoveries showed that restoring healthy gut microbiota composition or reversing irregular gut microbiota may benefit the disease’s remit and has minimal side effects [7,8]. New treatments that can target the gut microbiota to change the microbiota composition might be an alternative, independent, or complementary means to the traditional treatment [4,9]. Current microbiota-based treatment includes fecal microbiota transplantation (FMT) and drug-induced compositional change [10,11]. However, these methods require a long treatment time and often give delayed therapeutic efficacy. There is an urgent need for a drug delivery platform that can rapidly change microbiota composition while not generating severe side effects to the gut microenvironment of the host.

Studies indicated that dysbiosis of gut microbiota, usually a decreased Firmicutes/Bacteroidetes (F/B) ratio, is associated with IBD patients [12]. A medication that can quickly reverse the F/B ratio may have a therapeutic potential in the clinical treatment of IBD. We noticed that enteric bacteria could quickly uptake exosomal lipid nanoparticles (eLNP) derived from ginger [13], and 6-shogaol (6S)-containing ginger juice had substantial effects on the change of composition and function of gut microbiota in healthy people (after seven days of treatment) [14]. 6S is one of the major active compounds of ginger and is proven to have excellent antioxidative, anti-inflammatory, and anticarcinogenic properties [15,16,17]; also, poly-lactic acid nanoparticles (PLA-NPs)-delivered 6S has shown superior therapeutic efficacy against UC [18]. As our preliminary data indicated that 6S presented an effect on gut microbiota composition change with the potential of gradually increasing the F/B ratio, we hypothesized that 6S delivered by reassembled LNP from ginger lipids (6S/nLNP) could be readily uptaken by the enteric bacteria, and the natural LNP (nLNP) encapsulation might reduce the time required for the onset of action of 6S on compositional change of microbiota.

To test the hypothesis, we first study whether the common enteric bacteria *E. coli* could uptake reassembled 6S/nLNP, and found that 6S/nLNP, but not synthetic 6S-loaded poly-lactic acid nanoparticles (6S/PLA NPs), could quickly be taken up by *E. coli*. We then compare the 6S/nLNP with free 6S for their compositional modulation ability against the gut microbiota. We confirmed that the microbiota-targeting ability of nLNP is conducive to significantly accelerating the microbiota compositional change of 6S. Next, we investigate whether such compositional change could trigger downstream metabolic profile changes of the gut microbiota and does this metabolic profile alteration have a consequent functional contribution to the anti-inflammatory effect of 6S/nLNP.

## 2. Materials and Methods

### 2.1. Chemicals

6-shogaol, carboxymethylcellulose sodium salt (CMC-Na, medium viscosity), polyvinyl alcohol (PVA), polylactic acid (PLA), potassium chloride (KCl), ethanol (200 proof), dichloromethane (DCM), acetone, and acetonitrile (LC-MS grade) were purchased from Sigma-Aldrich (St. Louis, MO, USA). Formic acid (98%, LC-MS grade), phosphate buffer saline (Corning^TM^ PBS, 1×), and Dil Stain (1,1′-Dioctadecyl-3,3,3′,3′-Tetramethylindocarbocyanine Perchlorate (DiIC18(3))) were obtained from Fisher Scientific (Hampton, NH, USA). Fetal bovine serum (FBS) was obtained from R&D Systems (Flowery Branch, GA, USA). Dextran sodium sulfate (DSS; molecular weight 42 kDa) was obtained from ICN Biochemicals (Aurora, OH, USA). Ultrapure deionized water was supplied by a Milli-Q water system (Millipore, Bedford, MA, USA).

### 2.2. Preparation of 6S/nLNP and 6S/PLA NP

#### 2.2.1. Lipid Extraction

Fresh ginger (*Zingiber officinale* Rosc.) roots (~5.0 kg) were purchased from the Buford Highway farmers’ market (Doraville, GA, USA). Ginger-derived-nanoparticles band 2 (GDNPs-2; ~500 mg) was obtained using the ultracentrifugation method reported in our previous papers [19,20]. Total lipids extraction was extracted using a modified liquid-liquid extraction method reported by Bligh and Dyer. In general, 60 mL of methanol/dichloromethane with a volume ratio of 2:1 (*v*/*v*) was added to 15 mL of GDNPs-2 suspension (1 mg/mL in PBS) in a glass separatory funnel. After mixing, dichloromethane (20 mL) and ddH_2_O (20 mL) were added sequentially into the funnel. The mixture was shaken thoroughly at room temperature (RT) 10–15 times, and, after standing for 15 min, the bottom phase (organic phase) was separated and transferred into a fresh glass separatory funnel. Then the sample was washed once with 5 mL KCl solution (1 mol/L) and once with 5 mL of deionized water. Finally, the organic phase was dried in a vacuum rotavapor at ~45 °C and stored at −20 °C.

#### 2.2.2. Reassembly of 6S/nLNP and Construction of 6S/PLA NPs

A total of 0.2 mL 6S (5.0 mg/mL in ethanol) was mixed with 2 mL total lipids (5.0 mg/mL in dichloromethane) and dried under reduced pressure to obtain a thin lipids-complex film. The dried 6S-lipids film was then suspended in 5 mL PBS buffer (1×, Fisher scientific). After a bath sonication for 5 min, along with proper pipetting (~50 times), another 5 mL of PBS buffer was added and sonicated for another 5 min. Finally, the suspension was passed through NanoSizer^TM^ liposome extruder (T&T Scientific Corporation, Knoxville, TN, USA) 20–25 times (at 55 °C) with a 200 nm polycarbonate membrane. LNP control was made by the identical protocol without the adding of 6S as described above. For the *E. coli* uptake experiment, LNPs were labeled with a lipophilic carbocyanine dye, DiIC18(3). Briefly, LNPs (~1 mg/mL in PBS) were treated with DiL (10 μM) for 30 min at RT, and the reaction mixture was subjected to ultracentrifugation to remove the free Dil. Labeled LNPs pellet was then reconstituted in 10 mL of PBS buffer by proper pipetting.

To prepare Dil-labeled 6S/PLA NPs, PLA (100 mg), Dil (50 µL in 1mg/mL Acetone), and 6S (1mL, 5.0 mg/mL in ethanol) were dissolved in 5 mL of DCM. An oil-in-water emulsion was formed by emulsifying the polymer solution in 20 mL of 2.5% *w*/*v* aqueous PVA solution using a probe sonicator (Branson S-450; Danbury, CT, USA) at 50% amplitude for 30 sec (3 times) over an ice bath. The emulsion was stirred overnight at RT, followed by 2 h of rotary evaporation under vacuum to remove the residual DCM. Nanoparticles were recovered by 2 h centrifugation at 10,000 rpm (remove the micro-sized particle), followed by 45 min ultracentrifugation (25,000 rpm, at 4 °C) and washed three times with deionized water. 6S/PLA NPs suspension was then lyophilized to obtain a dry powder.

Particle sizes and zeta-potentials of prepared obtained 6S/nLNP or 6S/PLA NPs were measured using a Malvern Zetasizer Nano ZS90 Apparatus (Malvern Instruments, Worcestershire, UK) at RT.

### 2.3. Uptake of 6S/nLNP by E. coli

*E. coli* (ATCC-25922-GFP; green) was cultured in the 2855 tryptic soy broth medium (BD 211825) with 100 mcg/mL Ampicillin. After 1 h of reviving, above mentioned medium was replaced by a medium in the presence of Dil-labeled 6S/nLNP (1 μg/mL), free Dil (1 μg/mL), or Dil-labeled 6S/PLA NPs (1 μg/mL), and cultured for 4 h at 37 °C. The culture suspensions were then centrifuged at 1000× *g*, and the pellet was washed with PBS twice and fixed in 4% PFA, followed by nucleic acid staining with DAPI (blue). Images were acquired by Olympus BX63 fluorescence microscopy with a 60× objective (with oil) and a DP26 digital camera. Merged channels were processed with cellSens Dimension software. red fluorescence was quantitated by ImageJ (*n* = 3, *p* < 0.0001 ****, NS: non-significant).

### 2.4. Microbiota Composition Analysis upon Oral Administration of 6S or 6S/nLNP

#### 2.4.1. Animal

Female C57BL/6J mice (7–8 weeks old) were purchased from the Jackson Laboratory (Bar Harbor, ME, USA). All animals were acclimatized for more than three days in the experimental area. Animal experiments were performed under Georgia State University guidelines covering the humane care and use of animals in research. All animal procedures used in this study were approved by the University Committee on Use and Care Animals at Georgia State University (IACUC, Protocol # A17044).

Mice were divided into four groups (*n* = 6): control, free 6S, empty nLNP, and 6S/nLNP group. Mice were marked and housed (3 per cage) in polypropylene cages and maintained in controlled environmental conditions with 12 h light and dark cycles. The temperature and humidity of the room were maintained between 22–23 °C and 30–70%, respectively. Mice had fasted for 4 h before dose, and water was provided ad libitum through the study period. Microbiota composition studies were performed following 5-day daily oral (PO) administration of control (autoclaved PBS), free 6S (5 mg/kg, 0.5% CMC-Na in autoclaved PBS), empty nLNP (1 mg/mL in autoclaved PBS), and 6S/nLNP (5 mg/kg, in autoclaved PBS). Feces were collected from the individual mouse before treatment and on days 1, 3, 5, 10, 15, 20, 30, respectively. Collected fecal samples were stored in 1.5 mL Eppendorf tubes (sterile) at −80 °C until further analysis.

#### 2.4.2. 16S rRNA Gene Sequencing

16S rRNA gene amplification and sequencing were done using the Illumina MiSeq technology following the protocol of Earth Microbiome Project (http://www.earthmicrobiome.org/emp-standard-protocols, accessed on 21 March 2018). RNAs were extracted from frozen extruded feces using a PowerSoil kit (Qiagen) with mechanical disruption (bead-beating). The 16S rRNA genes, region V4, were PCR amplified from each sample using a composite forward primer and a reverse primer containing a 12-base barcode, which was used to tag PCR products from respective samples. The forward primer is 515F 5′-*AAT GAT ACG GCG ACC ACC GAG ATC TAC AC*
**TAT GGT AAT T*GT***
GTG CCA GCM GCC GCG GTA A-3′. The sequence in italic is the 5′ Illumina adaptor B, and the sequence in bold is the primer pad. The sequence in both italic and bold (GT) is the primer linker. The underlined sequence is the conserved bacterial primer 515F. The reverse primer is 806R 5′-*CAA GCA GAA GAC GGC ATA CGA GAT* XXX XXX XXX XXX **AGT CAG TCA G*CC***
GGA CTA CHV GGG TWT CTA AT-3′. The sequence in italic is the 3′ reverse complement sequence of Illumina adaptor, the 12X sequence is the Golay barcode, the sequence in bold is the primer pad, and the italicized and bold sequence is the primer linker. The underlined sequence is the conserved bacterial primer 806R. PCR reactions consisted of Hot Master PCR mix (Five Prime), 0.2 mM of each primer, 10–100 ng template, and reaction conditions were 3 min at 95 °C, followed by 30 cycles of 45 s at 95 °C, 60 s at 50 °C, and 90 s at 72 °C on a Bio-Rad thermocycler. Four independent PCRs were performed for each sample, combined, purified with AMPure XP magnetic purification beads (Beckman Coulter Life Sciences, IN, USA), and products were visualized by gel electrophoresis. Products were then quantified (BIOTEK Fluorescence Spectrophotometer) using a Quant-iT PicoGreen dsDNA assay. A master DNA pool was generated from the purified products in equimolar ratios. The pooled products were quantified using a Quant-iT PicoGreen dsDNA assay and then sequenced using an Illumina MiSeq sequencer (paired-end reads, 2 × 250 base pairs) at the Cornell University, Ithaca, NY, USA.

#### 2.4.3. Microbiota Composition Data Analysis and Metagenome Prediction

The sequences were demultiplexed, quality filtered using the Quantitative Insights into Microbial Ecology (QIIME, version 1.8.0) software package, and forward and reverse Illumina reads were joined using the fastq-join method (https://expressionanalysis.github.io/ea-utils/, accessed on 21 April 2018). We used QIIME default parameters for quality filtering (reads truncated at first low-quality base and excluded if: (1) There were more than three consecutive low-quality base calls; (2) less than 75% of reading length was consecutive high-quality base calls; (3) at least one uncalled base was present; (4) more than 1.5 errors were present in the barcode; (5) any Phred qualities were below 20; or (6) the length was less than 75 bases). Sequences were assigned to OTUs using the UCLUST algorithm with a 97% threshold of pairwise identity and with the creation of new clusters with sequences that did not match the reference sequences. OTUs were taxonomically classified using the Greengenes reference database. A single representative sequence for each OTU was aligned, and a phylogenetic tree was built using FastTree. The phylogenetic tree was used for computing the unweighted UniFrac distances between samples. Rarefied OTU tables were used to compare the abundances of OTUs across samples. Principal coordinates analysis (PCoA) plots were used to assess the variation between the experimental group (beta diversity), phylogenetic beta diversity metrics are calculated based on UniFrac distance. Alpha diversity curves were determined for all samples using the determination of the number of observed species, and the OTU table was rarefied at various taxonomic levels using QIIME.

### 2.5. Fecal Metabolites Extraction

Feces from individual mice were collected and freeze-dried by a lyophilizer (Labconco, Kansas City, MO, USA). In general, 200 mg of feces was used to extract the metabolites by adding 1.8 mL 80% icy cold methanol. After 30 min of sonication (in an ice bath), all samples were incubated at −20 °C for 1 h. The samples were then vortexed for ~2 min and centrifuged at 12,000 rpm at 4 °C for 10 min. The supernatant of the extraction was centrifugated through the membrane filter with a molecular weight cutoff (MWCO) of 3500 Dalton to remove the macromolecules and then lyophilized overnight. For the LC-MS analysis, the lyophilized powder was reconstituted in 50 µL of 50% methanol, filtered through a 0.22 µm filter, and transferred to 2.0 mL HPLC injection vials. LC-MS quality control (QC) sample was made by mixing 6 different samples, and LC retention times of QC samples were used to evaluate the stability of the LC-MS system.

For in vitro test, the dried metabolites were reconstituted in the culture medium in a series of diluted concentrations and used to treat the macrophages or the Caco-2 BBE cells. For in vivo model, the dried metabolites were reconstituted in PBS and vortexed to form a suspension.

### 2.6. Fecal Metabolic Profile


#### 2.6.1. Ultra-High-Performance Liquid Chromatography Time-of-Flight Mass Spectrometry (UPLC-TOF-MS)

High-resolution accurate mass spectrometry (HRMS) data were acquired by UPLC-Q-TOF-MS, equipped with Agilent 1290 Infinity II LC system and Agilent Q-TOF 6545 mass spectrometer (Agilent Technologies, Santa Clara, CA, USA). The acquisition was performed with ESI-MS in non-targeted MS/MS mode. The UPLC system contained a Zobax C_18_ column (2.1 × 50 mm, 1.8 μm; Agilent Technologies, Santa Clara, CA, USA) for the peak separation. The mobile phase was composed of solvent A (0.1% formic acid, 5% acetonitrile, and 94.9% water) and solvent B (0.1% formic acid−99.9% acetonitrile) and was run in a gradient elution mode. The flow rate of the UPLC was set as 0.2 mL per min, and the temperature of the column was kept at 40 °C. The sample injection module was kept at 4 °C all the time.

The mass spectrometry was performed using the following conditions: nebulizer pressure, 35 PSIG; capillary voltage, 3500 V; fragmentor voltage, 175 V; drying gas flow, 12 L/min; drying gas temperature, 320 °C; sheath gas flow, 11 L/min; sheath gas temperature, 350 °C. The mass range was recorded from *m/z* 50 to 1700 Dalton. Data acquisition was performed using the MassHunter Workstation (Agilent, Santa Clara, CA, USA).

#### 2.6.2. Metabolic Profile and Identification of Metabolites


Acquired UPLC-HRMS data were converted to *.mzXML format by open-source software (ProteoWizard, Version: 3.0.18320, Palo Alto, CA, USA) and uploaded to the XCMS website (https://xcmsonline.scripps.edu, accessed on 21 November 2018). The uploaded HRMS data from two groups of mouse fecal metabolites (control and 6s/nLNP treated) were matched via Metlin online metabolomics database (https://metlin.scripps.edu, accessed on 21 December 2018), which contains over a million molecules ranging from lipids, steroids, plant and bacteria metabolites, small peptides, carbohydrates, exogenous drugs/metabolites, central carbon metabolites, and toxicants. The searching used a positive charged mass feature, including [M+H]^+^, [M+Na]^+^, [M-H_2_O+H]^+^, and [M-H_2_O+Na]^+^, and with an M/z accuracy no more than 3.0 ppm.

### 2.7. In Vitro Anti-Inflammatory Assay of Fecal Metabolites

Mouse macrophage cells (RAW 264.7, ATCC, Manassas, VA, USA) were used to test the fecal metabolites’ in vitro anti-inflammatory effects. The passage number of the macrophage cells was between 5 and 10. Briefly, ~1.0 × 10^5^ macrophage cells were seeded in 6-well plates (Corning Life Sciences, Tewksbury, MA, USA) and cultured with 2 mL of Dulbecco’s Modified Eagle Medium (DMEM, Corning Life Sciences, Tewksbury, MA, USA) that contains 10% fetal bovine serum. After the cell reached ~75% of confluence (judged by the percentage of the plate bottom area covered with cells), the culture medium was replaced by a new DMEM medium (with 10% FBS) with fecal metabolites extracted from a different group of mice (between day 15 and day 20), and incubated for another 12 h. After the treatment, the culture medium was removed, and cells were activated by adding diluted LPS in the medium (200 ng/mL of LPS, without FBS) and incubate for another 4 h at 37 °C.

For the mRNA expression level test, total RNA was extracted and purified from RAW 264.7 cells by RNeasy Mini Kit (Qiagen, Hilden, DE, USA). RNA extracted from the samples was used to generate complementary DNA (cDNA) with Maxima cDNA Synthesis Kit (Thermo Scientific, Waltham, MA, USA). Maxima SYBR Green/ROX qPCR Master Mix (Thermo Scientific, Waltham, MA, USA) was used to analyze the RNA expression of pro-inflammatory cytokines (including TNF-α, IL-1β, and IL-6) and anti-inflammatory cytokines (IL-10). Results were normalized with 36B4, the housekeeping gene. Following sense and anti-sense primers were used: TNF-α: 5′-AGG CTG CCC CGA CTA CGT-3′ (forward) and 5′-GAC TTT CTC CTG GTA TGA GAT AGC AAA-3′ (reverse); IL-1β, 5′-TCG CTC AGG GTC ACA AGA AA-3′ (forward) and 5′-CAT CAG AGG CAA GGA GGA AAA C-3′ (reverse); IL-6: 5′-ACA AGT CGG AGG CTT AAT TAC ACA T-3′ (forward) and 5′-TTG CCA TTG CAC AAC TCT TTT C-3′ (reverse); IL-10: 5′-GGT TGC CAA GCC TTA TCG GA-3′ (forward) and 5′-CTT CTC ACC CAG GGA ATT CA-3′ (reverse); and 36B4: 5′-TCC AGG CTT TGG GCA TCA-3′ (forward) and 5′-CTT TAT CAG CTG CAC ATC ACT CAG A-3′ (reverse).

### 2.8. Electric Cell-Substrate Impedance Sensing (ECIS) Assay

Colonic epithelium cells (Caco2-BBE1, ATCC, Manassas, VA, USA) were used to measure the fecal metabolites’ wound-healing effects. Approximately 2.5 × 10^5^ (per well) Caco-2-BBE1 cells were seeded in 8-well ECIS cultureware plates (8W10E, Applied Biophysics, NY, USA) and cultured in 500 µL/well volume of DMEM (supplemented with 10% FBS). The plates were connected to the ECIS system by being fixed on the ECIS array station. The station was then placed in a cell culture incubator with 37 °C temperature, 5% CO_2_, and 90% humidity. The signal recording of ECIS was 500 Hz single frequency. After the cells’ resistance signals reach the plateau (about 20–30 h), the cell culture medium was replaced by a medium with 70% Ethanol (negative control), fecal metabolites from normal mouse group, with or without 6S/nLNP treated fecal metabolites. Next, a high-frequency (40 kHz) current (1400 µA, duration 30 s) was immediately applied to the plate to generate a consistent wound to each well of cells. Different groups of cells experience a healing stage and, the healing phase resistance signals were recorded for 20 h. At the end of the experiment, different groups’ resistance values were exported in an excel format and visualized by Prism 8 (GraphPad, San Diego, CA, USA).

### 2.9. In Vivo Efficacy Study

#### 2.9.1. Animal

A mouse model of acute ulcerative colitis was induced by using 2% DSS (in drinking water) for 7 days. On day 8, 2% DSS was replaced by water, and mouse was gavage fed with 0.2 mL normal fecal metabolites suspension (equivalent to 0.2 g feces) or 6S/nLNP-treated fecal metabolites suspension (equivalent to 0.2 g feces) daily for another 7 days. Metabolites were extracted from feces collected from day 10 to day 20 (with or without 6S/nLNP treatment). For every 0.2 g feces, 2 mL 80% icy cold methanol was added, the fecal suspension was then sonicated on an ice bath for 30 min and submitted to centrifugation to remove the debris (7500 rpm, 10 min, 4 C). The supernatant was then passed through Amicon™ Ultra Centrifugal Filter Units (3000 MWCO) to remove large biomolecules. The dosage used for each mouse is equivalent to extraction from 0.2 g feces. The body weight of the individual mouse was recorded once every two days. On day 14, mice were euthanized and the and tissues were collected for analysis.

#### 2.9.2. H&E Staining

Mouse colons were fixed for ~48 h in 10% buffered formalin solution at RT and then embedded in paraffin. After deparaffinization, the colon sections (~10 µm thickness) were stained with hematoxylin/eosin (H&E) using a standard protocol [21]. Images were acquired using an Olympus microscope equipped with a Hamamatsu Digital Camera (DP-26).

### 2.10. Statistical Analysis

Significance was determined using either unpaired two-tailed Student’s *t*-test or one-way ANOVA and differences were noted as significant (* *p* < 0.05, ** *p* < 0.01, *** *p* < 0.001, and **** *p* < 0.0001).

## 3. Results

### 3.1. Enteric Bacteria Effectively Take up 6S/nLNP

Previous research has proved that enteric bacteria (such as *E. coli*) can uptake ginger-derived exosomal lipid nanoparticles (eLNPs) [13]; in our study, we extracted total lipids from these eLNPs, and use a thin-film hydration method to reassemble them into a natural LNP (nLNP) delivery system [20,22]. We first wanted to testify that 6S-loaded reassembled nLNP could also be readily uptaken by the *E. coli*. As a comparison, we constructed a PLA polymeric delivery platform to encapsulate 6S. Both hydrophobic delivery systems are labeled with hydrophobic dye Dil(C18(3)) and loaded with the 6S. After 4 h of incubation with GFP-expressing *E. coli*, we found that only 6S/nLNP-incubated *E. coli* presented strong red fluorescence (Figure 1a), while the free Dil and Dil-labeled PLA polymer-incubated *E. coli* did not show observable red fluorescence. Although 6S-loaded PLA shares similar particle size (both around 200 nm) and zeta potential (~−18 mV) with 6S/nLNP (Figure 1c), 6S/PLA NPs lack the targeting effect to the *E. coli*. This study proved that reassembled nLNP, but not the PLA NPs, could target the enteric bacteria.

### 3.2. Oral Administration of 6S/nLNPs Accelerates 6S-Induced Changes in Gut Microbiota Composition

Drinking shogaol-containing ginger juice has been shown to have substantial effects on the composition and function of gut microbiota in healthy people after seven days [14]. Our preliminary data also indicated that the hydrophobic drug candidate, 6S, induced changes in the composition of the mouse gut microbiota after more than five days of treatment (Figure 2b). Because 6S/nLNPs can near-instantaneously target enteric bacteria, we asked whether 6S/nLNPs could accelerate the onset of 6S action on compositional changes in the gut microbiota. To this end, we randomly divided mice into four groups (*n* = 6/group): control, empty nLNPs, free 6S, and 6S/nLNPs. Mice were administered PBS, 1 mg/mL nLNPs, 0.1 mg/mL 6S, or 6S/nLNPs (0.1 mg/mL 6S, 1 mg/mL nLNPs) daily by oral gavage for five consecutive days, and feces were collected on different days for extraction of RNA for microbiota analysis (Figure 2a). As shown in Figure 2b, we found that orally delivered 6S/nLNPs changed the microbiota composition distinctively faster than free 6S, causing changes in the microbiota composition after only 24 h compared with the control group, as indicated by principal coordinates analysis (PCoA) (Figure 2b, red box). By comparison, PCoA analysis showed that free 6S-induced changes in microbiota composition appeared between days five and 10. Moreover, the Firmicutes/Bacteroidetes (F/B) ratio of gut microbiota was shown to significantly increase at the phylum level after one day of 6S/nLNP treatment (Figure 2c) compared with that for free 6S and the control group, underscoring the distinction in kinetics between 6S/nLNPs effects and those of 6S compared with controls. A PCoA analysis of feces continuously collected for an additional 25 days after the five-day treatment showed that the compositional changes induced by 6S/nLNPs were maintained even after treatment had ceased (Figure 2b, day 10).

### 3.3. Changes in Microbiota Composition Produce Distinct Changes in Metabolic Profile

After confirming that 6S/nLNPs could accelerate 6S-induced changes in microbiota composition, we next investigated whether this compositional change resulted in different metabolic properties. Using a liquid chromatography-mass spectrometry (LC-MS)-based untargeted-metabolomics technique, we found that the fecal metabolic profile was significantly changed after a five-day 6S/nLNP treatment (Figure 3a,d). Similar to the case for microbiota compositional changes, which were maintained after day five (Figure 2b), LC-MS profiles of fecal samples extracted from day five and day 15 revealed that the changes in metabolic profiles were also maintained (Figure 3a, day five and day 15). As shown in Figure 3b, a cloud plot demonstrated that more MS features were significantly upregulated (green; 1590 of 1844) than downregulated (red; 254 of 1844) after treatment with 6S/nLNPs for five-days. Additionally, more new metabolites appeared (60) than disappeared (24) after 6S/nLNP treatment, as indicated in the Venn diagram (Figure 3c). Interestingly, many significantly upregulated metabolites (Figure 3d, heatmap, yellow) induced by 6S/nLNP treatment do not have matched mass features in currently available databases, including Metlin [23,24], KEGG [25], and the NP Atlas [26].

We further analyzed the chemical composition of fecal metabolites from samples before (day −1) and after (day 15) treatment, and tentatively identified significantly altered metabolites (Table 1) by matching their high-resolution mass spectra (HRMS) data with the Metlin database [24]. We found that several bacterial metabolites, such as bacteriocin 28b and undecylprodigiosin, were clearly upregulated after 6S/nLNP treatment. As bacteriocin 28b is mainly biosynthesized by *Serratia marcescens* [27,28,29] and undecylprodigiosin is produced by *Actinomycetes* [30], our finding suggested that alternations in gut microbiota metabolism might be associated with these specific bacteria.

### 3.4. Gut Microbiota Composition Can Be Restored

Next, we monitored the microbiota to determine whether the observed compositional changes induced by 6S/nLNP treatment are permanent or temporary, and if temporary, how long they are maintained. PCoA analyses (Figure 4a) showed that compositional changes in the microbiota of both 6S/nLNP and free 6S groups lasted until day 20, and returned to normal by day 30 in both groups. F/B ratios in both groups also returned to a range similar to that of the control group (Figure 4c,d).

Operational taxonomy units (OTUs) tests, which use mathematically-defined taxa to assess microbiota composition, showed no significant effect of 6S/nLNPs or free 6S treatment, indicating that neither treatment affected the diversity of the gut microbiota. Taken together, these data indicate that 6S or 6S/nLNP treatment has the ability to exert compositional changes, but each is safe and produces reversible changes with minimal effects on the diversity of gut microbiota.

### 3.5. Altered Fecal Metabolites Exert Anti-Inflammatory Effects In Vitro

Having established these advantages of nLNP delivery of 6S, we then tested if the changes in the composition of microbiota and metabolites make a functional contribution to the reported anti-inflammatory effect of 6S [18,20]. We first sought to rule out effects attributable to metabolites of 6S by using only metabolites derived from the compositionally changed microbiota. Since mice were treated for five days continuously, and nLNPs protect 6S against rapid metabolism caused by gut microbiota and the host, we waited another five days and tested feces samples on day 10 to determine whether 6S metabolites are eliminated. Extracted-ion chromatogram (EIC) LC-MS data showed that most 6S-derived metabolites could be detected in feces on day five, but were absent on day 10 (Appendix A, day 10). Accordingly, only feces collected from day 10 to day 20 were used for subsequent proof-of-concept efficacy studies.

The in vitro anti-inflammatory effects of these metabolites were then evaluated against macrophage cells using the RAW 264.7 cell line. To test whether compositional changes in metabolites could prevent LPS-induced inflammation in macrophages, we incubated RAW 264.7 macrophages with complete culture medium containing metabolites from normal feces (equivalent to 0.1 mg/mL feces) or from day 10 6S/nLNP-treated feces. After 12 h, the metabolites-containing medium was replaced with serum-free medium, and LPS (final concentration, 200 ng/mL) was added to cultures to activate the macrophages. As shown in Figure 5a–d, treatment with microbiota-derived metabolites from 6S/nLNP-treated mice significantly decreased the levels of pro-inflammatory cytokines (TNF-α, IL-1β, and IL-6) but did not appear to increase the mRNA expression of the anti-inflammatory cytokine, IL-10, compared with control (normal fecal metabolites) or LPS-only groups. These results indicate that microbiota-derived metabolites from 6S/nLNP-treated mice exert an anti-inflammatory effect against LPS-induced inflammation and that this effect may not involve IL-10–dependent pathways.

Next, we tested the mucosal wound-healing effects of microbiota-derived metabolites from 6S/nLNP-treated mice, given that such an action would synergistically enhance the anti-inflammatory effects and favor the recovery of UC patients. The epithelial wound-healing ability of microbiota-derived metabolites from 6S/nLNP-treated mice was first tested in vitro using the electric cell-substrate impedance sensing (ECIS) method. Caco2-BBE cells cultured to confluence on ECIS 8W1E plates were subjected to a voltage pulse with a frequency of 40 kHz, amplitude of 4.5 V, and duration of 30 s [31]. This process kills the cells around the small active electrode, causing their detachment and generating a wound that is normally healable by surrounding cells that have not been affected by the voltage pulse. Wound healing was then assessed by measure resistance continuously for ~30 h after the wound. As shown in Figure 5e, the epithelial layer of wounded cells treated with microbiota-derived metabolites from 6S/nLNP-treated mice showed significantly accelerated recovery during the incubation period (~50 h) compared with those of the blank (medium), control (normal fecal metabolites), and negative control (70% ethanol) groups. These results suggest that microbiota-derived metabolites from 6S/nLNP-treated mice promote epithelium wound healing in vitro.

### 3.6. In Vivo Efficacy of Metabolites Derived from 6S/nLNP-Induced Changes in the Microbiota

Given the excellent in vivo anti-inflammatory efficacy of nLNP-delivered 6S reported by previous studies [18,20], and our demonstration that microbiota-derived metabolites obtained following 6S treatment showed beneficial effects in vitro, we tested whether these metabolites also play a crucial role in the in vivo efficacy of 6S. Initially, mice were provided 2% dextran sulfate sodium (DSS) in drinking water for seven days to induce an intestinal wound. In the following seven days, mice were given a once-daily oral dose of normal microbiota-derived metabolites or microbiota-derived metabolites from 6S/nLNP-treated mice (equivalent to 0.2 mg feces per dose) by gavage (Figure 6a). The control group provided water for the entire experimental period showed a slight increase in body weight (Figure 6b), but DSS-wounded mice showed a decrease in body weight during the wounding phase. After day eight, mice in groups treated with normal microbiota-derived metabolites or microbiota-derived metabolites from 6S/nLNP-treated mice showed increases in body weight, but the latter group gained body weight much faster than the former group. By day 14, mice in the group treated with microbiota-derived metabolites from 6S/nLNP-treated mice recovered more of their original body weight than those in the group treated with normal microbiota-derived metabolites. Measurements of colon length (Figure 6c) also showed a longer average colon length in this group than the group treated with normal microbiota-derived metabolites, indicating a better healing-promoting effect of microbiota-derived metabolites from 6S/nLNP-treated mice.

Similarly, a histological analysis (Figure 6d) confirmed that intestinal mucosal ulceration was significantly decreased in mice treated with microbiota-derived metabolites from 6S/nLNP-treated mice, whereas some mucosal ulcerations were clearly present in the group treated with normal microbiota-derived metabolites (Figure 6d, white arrows in middle column). We also found that mRNAs for the pro-inflammatory cytokines, TNF-α and IL-6, were slightly higher in the group treated with normal microbiota-derived metabolites compared with the group treated with microbiota-derived metabolites from 6S/nLNP-treated mice (Figure 6e). The ability to decrease mRNA expression levels of TNF-α and IL-6 in colon tissue may explain the efficacy of microbiota-derived metabolites from 6S/nLNP-treated mice in accelerating the healing of intestinal mucosal injuries.

## 4. Discussion

Microbiota-based therapies can potentially correct the dysbiosis that drives the dysregulated immune response in UC and are suggested to be safer than traditional approaches [10,32,33]. However, current microbiota-based therapies such as FMT, prebiotics, and probiotics transplant require a relatively long treatment time to modulate the gut microbiota composition [32,34]. FMT and biotics transplants are also associated with potential risks, such as pathogen transfer and increased obesity [35,36,37]. Thus, there is a need to develop more appealing strategies.

Plant-derived exosomal nanoparticles appear to be a promising and safe platform for colon-targeted drug delivery [38]. These exosomal nanoparticles are naturally occurring LNPs with lipid bilayer membranes. Lipids isolated from ginger-derived exosomal nanoparticles are mainly composed of phosphatidic acid (PA), monogalactosyl diacylglycerol (MGDG) and digalactosyl diacylglycerol (DGDG) [19,38], the latter two of which are glycerol lipids containing a polar galactosyl head and non-polar fatty acid tail. Galactose is known to be an essential factor for the survival and virulence of bacteria [39,40]. For example, *E. coli* takes up galactose and utilizes it in the Leloir pathway for glycogenesis or synthesis of mucopolysaccharide and glycoprotein. The presence of a galactosyl moiety in ginger nLNPs suggests the possibility that nLNPs can naturally target the gut microbiota. In contrast, although polymeric NPs, such as PLA, share similar particle size (both around 200 nm) and zeta potential (~−18 mV) with nLNP, they lack an enteric bacteria-targeting effect.

Our strategy of using nLNP-delivered drugs—or even metabolites from the compositionally changed gut microbiota of a normal healthy individual—has real-world translational potential. Mouse models are widely used in gut microbiota compositional change studies [41]; our current study is a proof-of-principle study in which nLNP-delivered 6S demonstrated the ability to almost instantly change the composition of the mouse gut microbiota, which in turn created functional changes in the anti-inflammatory fecal metabolome. The near-instant targeting ability of nLNPs expedites the compositional change in the gut microbiota—a crucial feature in the treatment of UC because delaying UC treatment may increase the likelihood of developing colitis-associated colon cancer. Both nLNP-delivered drug and drug-modified microbiota metabolites could be used as potential approaches for the treatment of UC. As the mouse model has its specificity/limitation, further tests of nLNP on other models are required before we can move on to explore the nLNP’s translational potential. Future applications of our current findings may also include using nLNP to deliver other therapeutics that instantly change the gut microbiota compositions.

One limitation of our study is that the *E. coli* abundance was not considered for nanoparticles uptake experiments. Additionally, the specific metabolites that exert anti-inflammatory consequences remain undefined, an issue that will require further research. Our study also suggests that 6S/nLNP-induced compositional changes are reversible. Although this characteristic of nLNP therapy enhances its biocompatibility, it also needs to be considered in the context of optimizing dosing strategy for long-term treatment based on the severity of the disease.

## 5. Conclusions

Natural LNPs, reassembled from ginger lipids, constitute an excellent drug-delivery platform that can target the gut microbiota. Natural LNP-delivered 6S accelerates the process of 6S-induced microbiota compositional changes in the F/B ratio at the phylum level. Such changes in the composition of the microbiota trigger downstream metabolic changes in feces that exert beneficial anti-inflammatory effects—both in vitro and in vivo—that are conducive to modulating UC. Our study offers a potential approach for managing UC through near-instantaneous regulation of gut microbiota, both compositionally and functionally.

## Figures and Tables

**Figure 1 pharmaceutics-13-01355-f001:**
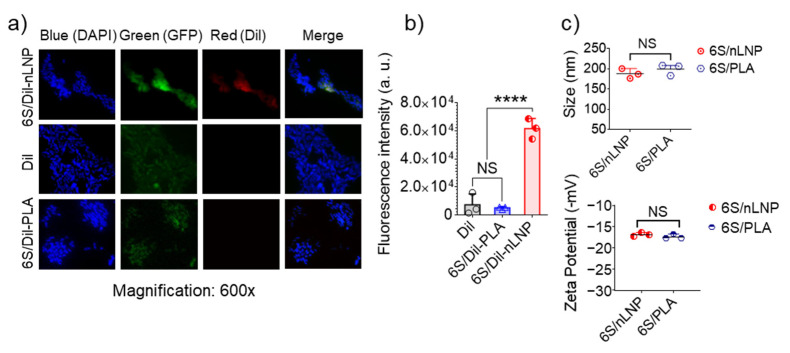
*E. coli* uptake of reassembled lipid NPs. *E. coli* (ATCC-25922-GFP; green) were cultured in the presence of Dil-labeled 6S/nLNPs (1 μg/mL), free Dil (1 μg/mL), or Dil-labeled 6S/PLA NPs (1 μg/mL) for 4 h, washed with PBS and fixed in 4% PFA, followed by staining of nuclei with DAPI (blue). Images were acquired using an Olympus BX63 fluorescence microscope with a 60× objective. Data are representative of three experiments. (**a**) Representative microscopic images of *E. coli* treated with Dil-labeled 6S/nLNPs (6S/Dil-nLNP), free Dil, or Dil-labeled 6S/PLA NPs (6S/Dil-PLA). (**b**) Comparison of red fluorescence intensity quantified using ImageJ (*n* = 3; **** *p* < 0.0001, NS: non-significant). (**c**) Comparison of size and zeta potential of 6S/nLNPs and 6S/PLA NPs (*n* = 3, NS: non-significant).

**Figure 2 pharmaceutics-13-01355-f002:**
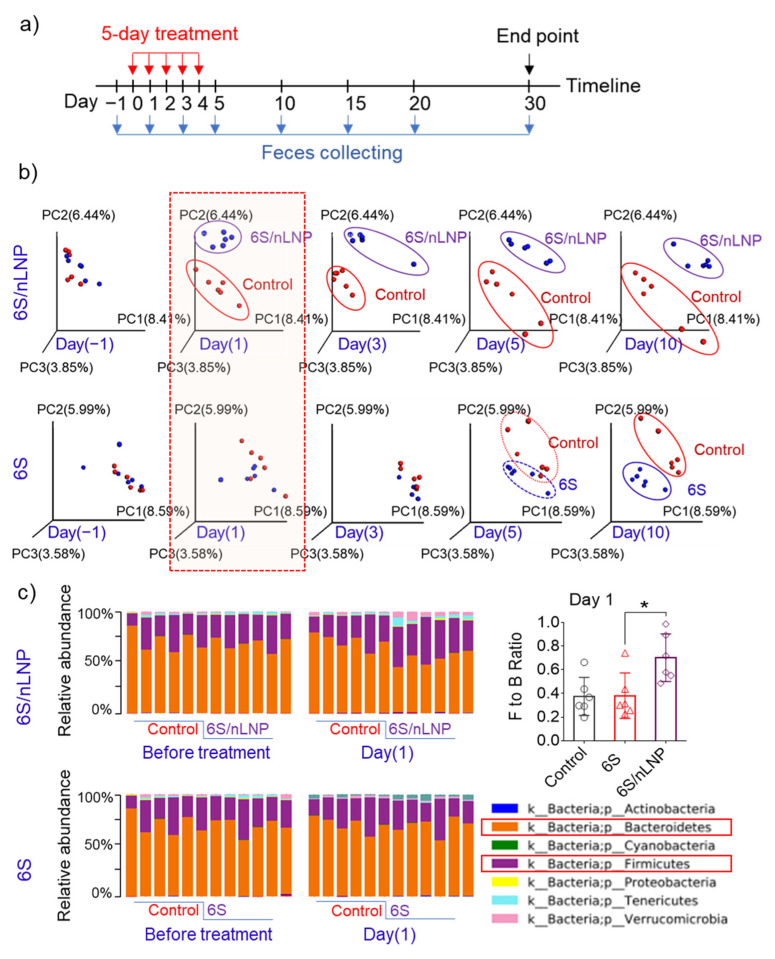
The onset of action of changes in microbiota composition is much faster in mice treated with 6S/nLNP than in those treated with free 6S. (**a**) Schematic diagram of treatment and feces collection. (**b**) PCoA of microbiota composition in mice treated with 6S/nLNPs or free 6S relative to a control group of mice (days −1, 1, 3, 5, and 10). (**c**) Taxonomic summary at the phylum level and Firmicutes/Bacteroidetes (F/B) ratio of gut microbiota (day 1) with 6S/nLNP or free 6S versus control treatment (*n* = 6; * *p* < 0.05).

**Figure 3 pharmaceutics-13-01355-f003:**
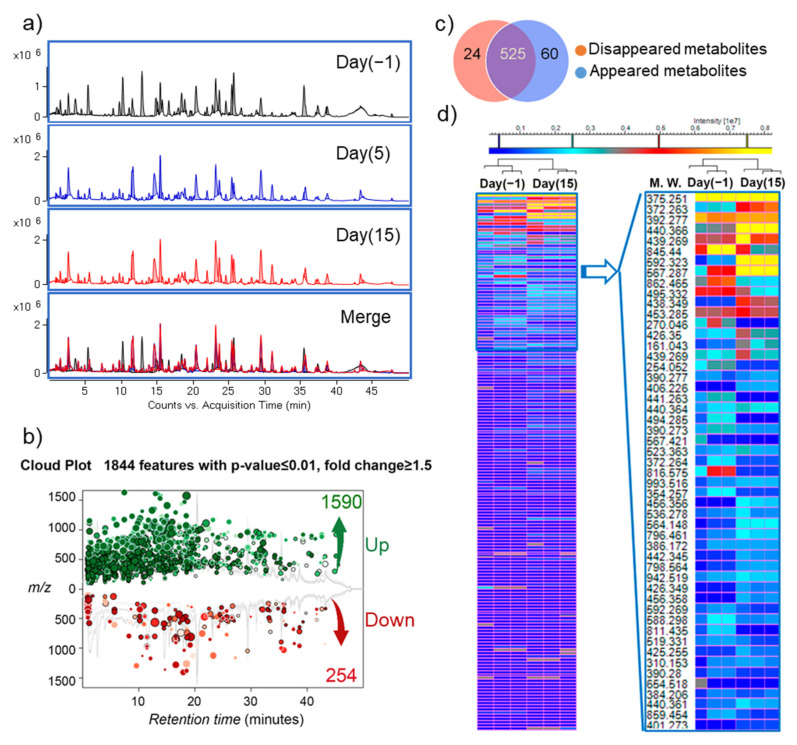
6S/nLNP treatment alters fecal metabolites. (**a**) Representative UPLC-MS (ESI+, Q-TOF) total ion current (TIC) profiles of fecal metabolites before (day 0) and after (day 5, day 15) 6S/nLNP treatment. (**b**) Cloud plot demonstrating upregulated MS features (green; 1590 of 1844) and downregulated MS features (red; 254 of 1844) after five days of 6S/nLNP treatment (day 15 sample). (**c**) Venn diagram of disappeared or appeared fecal metabolites before and after 6S/nLNP treatment, as detected by LC-MS. Twenty-four metabolites disappeared and 60 new metabolites appeared after 5 days of treatment (day 15 sample). (**d**) Heatmap summary of the numerous significantly upregulated metabolites on day 15 induced by a five-day 6S/nLNP treatment and their possible molecular weights (MW.).

**Figure 4 pharmaceutics-13-01355-f004:**
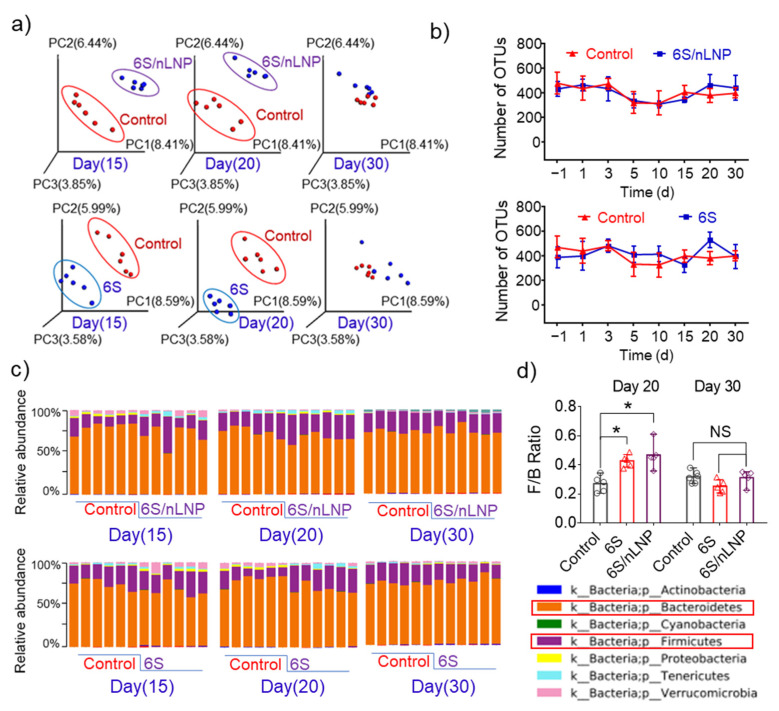
The diversity of microbiota is not affected by free 6S or 6S/nLNP treatment. (**a**) PCoA analysis of microbiota in mice treated with 6S/nLNP or free 6S relative to control mice (days 15, 20, and 30). (**b**) The number of operational taxonomy units (OTUs) in the microbiota of mice treated with 6S/nLNP or free 6S compared with that in the control group. (**c**) Taxonomic summary at the phylum level (days 15, 20, and 30), and (**d**) F/B ratio of gut microbiota (days 20 and 30) for mice treated with 6S/nLNPs or free 6S versus controls (*n* = 6; * *p* < 0.05, NS: non-significant).

**Figure 5 pharmaceutics-13-01355-f005:**
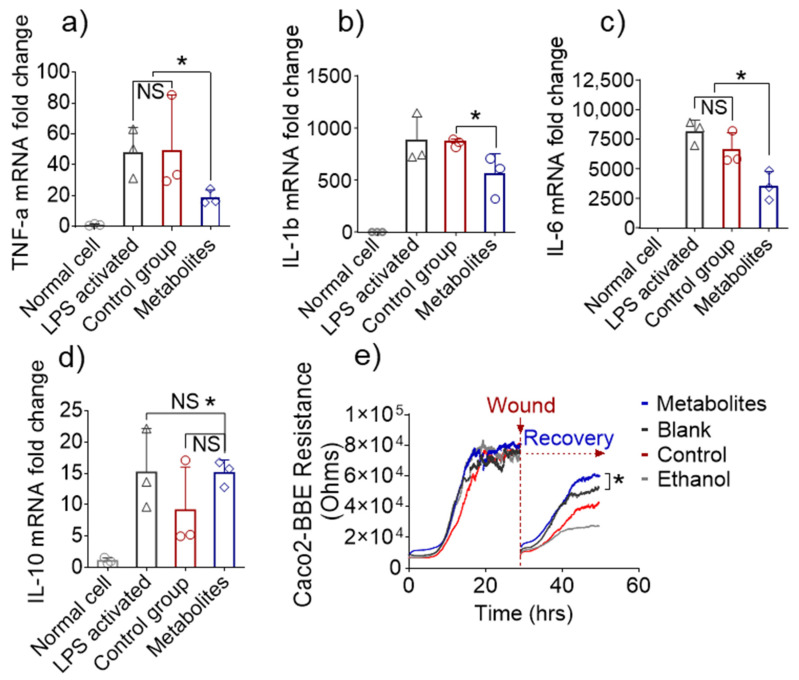
In vitro anti-inflammatory and wound-healing effects. (**a**) Fold-changes in the expression of TNF-α, (**b**) IL-1β, (**c**) IL-6, and (**d**) IL-10 genes in inflamed Raw 264.7 cells treated with microbiota-derived metabolites from 6S/nLNP-treated mice or normal microbiota-derived metabolites (*n* = 3; * *p* < 0.05 (**a**–**d**), NS: non-significant). (**e**) Representative graph showing the effects of microbiota-derived metabolites from 6S/nLNP-treated mice or normal microbiota-derived metabolites on the healing of wounded Caco-2 BBE cells (*n* = 3; * *p* < 0.05).

**Figure 6 pharmaceutics-13-01355-f006:**
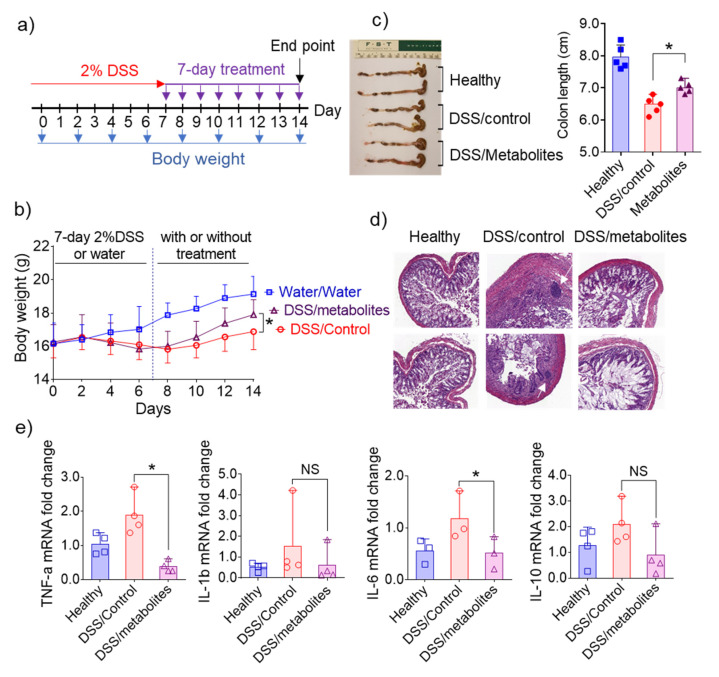
Effects of microbiota-derived metabolites from 6S/nLNP-treated mice in an in vivo wound-healing model. (**a**) Schematic diagram showing treatment and feces collection. (**b**) Body weight changes in control mice (water only; gray) and mice treated for seven days with 2% DSS, followed by seven days of microbiota-derived metabolites from 6S/nLNP-treated mice (purple) or seven days of normal microbiota-derived metabolites (red) (*n* = 5; * *p* < 0.05). (**c**) Colon lengths in each group of mice (*n* = 5; * *p* < 0.05). (**d**) H&E-staining. White arrows indicate inflammatory cells in the lamina propria. (**e**) Measurement of mRNA levels of cytokines in the DSS-induced mouse model of wound healing (*n* = 4; * *p* < 0.05, NS: non-significant).

**Table 1 pharmaceutics-13-01355-t001:** Part of tentatively identified major fecal metabolites before and after 6S/nLNP treatment.

Name	Formula	CAS	METLIN ID	Mass	ΔPPM	After Treatment
Bacteriocin 28b	C_12_H_11_N_3_O_2_	148466-49-3	89961	229.0851	0	↑
Undecylprodigiosin	C_25_H_35_N_3_O	52340-48-4	69260	393.2780	1	↑
NAc-DNP-Cys	C_11_H_11_N_3_O_7_S	35897-25-7	69126	329.0318	4	↑
Iprodione	C_13_H_13_Cl_2_N_3_O_3_	36734-19-7	68915	329.0334	0	↑
24 unknown metabolites	n/a	n/a	n/a	n/a	n/a	↓
60 unknown metabolites	n/a	n/a	n/a	n/a	n/a	↑

## Data Availability

The data underlying this article will be shared at reasonable request to the corresponding author.

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
