# Peer review of "Orally Administered Natural Lipid Nanoparticle-Loaded 6-Shogaol Shapes the Anti-Inflammatory Microbiota and Metabolome"

_pharmaceutics, 2021, doi:10.3390/pharmaceutics13091355_

Round 1
Reviewer 1 Report
I find the study interesting, with a clear structure and proper methods and well explain results. I suggest reinforcing the introduction to introduce better some concepts related to the use of 6-shogaol and better explain why the study is necessary. Also, I suggest including some aspects in the discussion section to give strength to the study and emphasise in the future application of the results.
Introduction:
The authors state a change in microbiota in Ulcerative colitis, but they do not provide information about specific changes produced by the disease and how they expect that treatment will reverse these changes.
There is little information about 6-shogaol? Why did they decide to use this compound? Is it a drug used in UC treatment? Is it an anti-inflammatory drug? May authors briefly explain the importance of this compound in the introduction section.
Methods:
Unifrac distance was used to calculate beta-diversity? Please clarify how beta-diversity was calculated?
Results:
Lines 365-366 elaborate better on the relation between metabolites and mentioned bacteria.
Discussion:
The abundance of E. coli may influence the uptake of 6S/nLNP, are other enteric bacteria besides E. coli able to uptake the compound?. It would be interesting to discuss these aspects in the discussion section or add as a limitation of the study if E. coli abundance was not considered for experiments.
In the discussion section, it may be worth elaborating on the use of mouse models and the specificity of results in this model because microbiota for other models may act distinctly.
Authors could elaborate on the future applications of the results
Comment to authors:
The green genes database is not updated since 2013. For further studies, I suggest using SILVA or RDP databases which includes new sequences.
Author Response
Reviewer #1
I find the study interesting, with a clear structure and proper methods and well explain results. I suggest reinforcing the introduction to introduce better some concepts related to the use of 6-shogaol and better explain why the study is necessary. Also, I suggest including some aspects in the discussion section to give strength to the study and emphasize in the future application of the results.
Introduction:
The authors state a change in microbiota in Ulcerative colitis, but they do not provide information about specific changes produced by the disease and how they expect that treatment will reverse these changes.
Response: Thanks for raising an important point related to treatment expectations. Firmicutes and Bacteroidetes are the two most critical bacterial phyla in the gastrointestinal tract, and the Firmicutes/Bacteroidetes (F/B) ratio has an important influence in maintaining normal intestinal homeostasis. Increased or decreased F/B ratio is regarded as dysbiosis, whereby the former is usually observed with obesity and the latter with inflammatory bowel disease (IBD) [1]. We expect that our nLNP-loaded 6-shogaol could reverse the decrease of the F/B ratio observed in the ulcerative colitic mice. We have now added the specific changes produced by the disease and our treatment expectations in the introduction.
[1] Stojanov S, Berlec A, Štrukelj B. The Influence of Probiotics on the Firmicutes/Bacteroidetes Ratio in the Treatment of Obesity and Inflammatory Bowel disease. Microorganisms. 2020 Nov 1;8(11):1715. doi: 10.3390/microorganisms8111715. PMID: 33139627; PMCID: PMC7692443.
There is little information about 6-shogaol? Why did they decide to use this compound? Is it a drug used in UC treatment? Is it an anti-inflammatory drug? May authors briefly explain the importance of this compound in the introduction section.
Response: Thanks for pointing this out to us. 6-shogaol is reported to have excellent antioxidative, anti-inflammatory, and anticarcinogenic properties [2,3,4], it also shows superior therapeutic efficacy against UC in our previous study [5]. Thus, we decide to use 6-shogaol to test whether it can modulate the composition of gut microbiota. We have now explained the importance of 6-shogaol in the introduction section.
[2] Nonaka K, Bando M, Sakamoto E, Inagaki Y, Naruishi K, Yumoto H, Kido JI. 6-Shogaol Inhibits Advanced Glycation End-Products-Induced IL-6 and ICAM-1 Expression by Regulating Oxidative Responses in Human Gingival Fibroblasts. Molecules. 2019 Oct 15;24(20):3705. doi: 10.3390/molecules24203705. PMID: 31619000; PMCID: PMC6832546.
[3] Bischoff-Kont I, Fürst R. Benefits of Ginger and Its Constituent 6-Shogaol in Inhibiting Inflammatory Processes. Pharmaceuticals (Basel). 2021 Jun 15;14(6):571. doi: 10.3390/ph14060571. PMID: 34203813; PMCID: PMC8232759.
[4] Pan MH, Hsieh MC, Kuo JM, Lai CS, Wu H, Sang S, Ho CT. 6-Shogaol induces apoptosis in human colorectal carcinoma cells via ROS production, caspase activation, and GADD 153 expression. Mol Nutr Food Res. 2008 May;52(5):527-37. doi: 10.1002/mnfr.200700157. PMID: 18384088.
[5] Zhang M, Xu C, Liu D, Han MK, Wang L, Merlin D. Oral Delivery of Nanoparticles Loaded With Ginger Active Compound, 6-Shogaol, Attenuates Ulcerative Colitis and Promotes Wound Healing in a Murine Model of Ulcerative Colitis. J Crohns Colitis. 2018 Jan 24;12(2):217-229. doi: 10.1093/ecco-jcc/jjx115. PMID: 28961808; PMCID: PMC5881712.
Methods:
Unifrac distance was used to calculate beta-diversity? Please clarify how beta-diversity was calculated?
Response: Phylogenetic beta diversity metrics are calculated based on UniFrac distance. We have now added this in the method section.
Results:
Lines 365-366 elaborate better on the relation between metabolites and mentioned bacteria.
Response: We appreciate the reviewer’s comment to elaborate better on the relation between metabolites and mentioned bacteria. We have tentatively identified fecal metabolites bacteriocin 28b and undecylprodigiosin by LC-MS; according to literature, bacteriocin 28b is mainly biosynthesized by Serratia marcescens [6,7], and undecylprodigiosin is produced by Actinomycetes [8]. We have now added the information to the results section.
[6] Donia, M.S.; Fischbach, M.A. HUMAN MICROBIOTA. Small molecules from the human microbiota. Science 2015, 349, 1254766, doi:10.1126/science.1254766.
[7] Guasch, J.F.; Enfedaque, J.; Ferrer, S.; Gargallo, D.; Regue, M. Bacteriocin 28b, a chromosomally encoded bacteriocin produced by most Serratia marcescens biotypes. Res Microbiol 1995, 146, 477-483, doi:10.1016/0923-2508(96)80293-2.
[8] Abdelmohsen, U.R.; Grkovic, T.; Balasubramanian, S.; Kamel, M.S.; Quinn, R.J.; Hentschel, U. Elicitation of secondary metabolism in actinomycetes. Biotechnol Adv 2015, 33, 798-811, doi:10.1016/j.biotechadv.2015.06.003.
Discussion:
The abundance of E. coli may influence the uptake of 6S/nLNP, are other enteric bacteria besides E. coli able to uptake the compound? It would be interesting to discuss these aspects in the discussion section or add as a limitation of the study if E. coli abundance was not considered for experiments.
Response: We agree with the reviewer’s suggestion and have now added a sentence stating that “One limitation of our study is that the E. coli abundance was not considered for nanoparticles uptake experiments”.
In the discussion section, it may be worth elaborating on the use of mouse models and the specificity of results in this model because microbiota for other models may act distinctly.
Response: Mouse models are widely used in gut microbiota compositional change studies [9]; our current study is a proof-of-principle study that nLNP delivered drug could have a near-instant effect on the compositional and functional changes of mouse gut microbiota. We fully agree with the reviewer that microbiota for other models may act distinctly; further preclinical research on other models is required before we can move on to explore the nLNP’s translational potential.
[9] Hugenholtz F, de Vos WM. Mouse models for human intestinal microbiota research: a critical evaluation. Cell Mol Life Sci. 2018;75(1):149-160. doi:10.1007/s00018-017-2693-8
Authors could elaborate on the future applications of the results
Response: The future application of our current findings may also include using nLNP to deliver other therapeutics that instantly change the gut microbiota compositions.
Comment to authors:
The green genes database is not updated since 2013. For further studies, I suggest using SILVA or RDP databases which includes new sequences.
Response: Thanks for the reviewer’s suggestion; we will use SILVA or RDP in future studies.
Reviewer 2 Report
The submitted manuscript is very well written and has the potential of disseminating up-to-date information on potentially new treatments of ulcerative colitis, disease for which the unmet medical need still exists. The approach is timely and relevant and deserves attention of the basic and preclinical researchers. The concept describing near-instantly modulating microbiota composition and function with biologically active compound formulated in nanoparticle(s) could be developed further along with the proper drug development process . The scientific language is of high quality. I strongly advise the publication of the paper in its present form.
Author Response
The submitted manuscript is very well written and has the potential of disseminating up-to-date information on potentially new treatments of ulcerative colitis, disease for which the unmet medical need still exists. The approach is timely and relevant and deserves attention of the basic and preclinical researchers. The concept describing near-instantly modulating microbiota composition and function with biologically active compound formulated in nanoparticle(s) could be developed further along with the proper drug development process. The scientific language is of high quality. I strongly advise the publication of the paper in its present form.
Response: Thanks for the reviewer’s encouraging comments.